# Chemosensitivity of Lung Metastatic High-Grade Synovial Sarcoma

**DOI:** 10.3390/jcm10245956

**Published:** 2021-12-18

**Authors:** Cecilia Tetta, Grazia Montrone, Alessandra Longhi, Michele Rocca, Francesco Londero, Gianmarco Parise, Orlando Parise, Jos G. Maessen, Marco Miceli, Sandro Gelsomino

**Affiliations:** 1IRCCS Istituto Ortopedico Rizzoli, 40136 Bologna, Italy; cecilia.tetta@ior.it (C.T.); alessandra.longhi@ior.it (A.L.); michele.rocca@ior.it (M.R.); marco.miceli@ior.it (M.M.); 2Radiology Unit, S. Orsola-Malpighi Hospital, Alma Mater Studiorum—University of Bologna, 40138 Bologna, Italy; gramontrone@gmail.com; 3Cardiovascular Research Institute Maastricht—ARIM, Maastricht University Medical Center, 6229 ER Maastricht, The Netherlands; francesco.londero@asufc.sanita.fvg.it (F.L.); g.parise@maastrichtuniversity.nl (G.P.); oparise@libero.it (O.P.); j.g.maessen@mumc.nl (J.G.M.)

**Keywords:** synovial sarcoma, chemotherapy, lung metastases, lung metastasectomy, soft tissue sarcoma

## Abstract

*Background:* Synovial sarcoma is a relatively chemosensitive type of soft tissue sarcoma and it often metastasizes to the lung. We investigated the role of adjuvant chemotherapy in patients with high-grade synovial sarcoma at their first lung metastasectomy (LMTS). *Methods:* Forty-six HGSS patients had their first LMTS at our institute (Rizzoli Orthopedic Hospital, Bologna, Italy) between 2000 and 2020. We divided them into two groups: (1) those undergoing adjuvant chemotherapy (*n* = 24) and (2) those not receiving adjuvant chemotherapy (*n* = 22). The primary outcome was a median survival at 32.5 (IQR 18.0–82.7) median follow-up. The disease-free interval was calculated at time zero (DFI_0_, interval between the diagnosis of the primary tumor and the first CT-diagnosed lung metastasis) and at any further lung relapse (DFI_1–3_). T_0_ was defined as the time at first LMTS and T_1_–T_3_ referred to the time of further metastasectomy. *Results:* Freedom from SS-specific mortality at 60 months was significantly higher in patients without chemotherapy (50.0% (33.0–76.0%) vs. 20.8% (9.55%–45.4%), *p* = 0.01). Chemotherapy was associated with a higher risk of SS-specific mortality at multivariable Cox regression (HR 2.8, *p* = 0.02). Furthermore, DFI_0_ ≤ 6 months, female sex, age > 40 years, and primary tumor > 10 cm increased the risk of death by about four, six, >three, and >five times, respectively. *Conclusions.* Adjuvant chemotherapy did not show any advantage in terms of freedom from SS-specific mortality in HGSS patients. Further larger studies are necessary to confirm our findings.

## 1. Introduction

Synovial sarcoma (SS) is a rare, highly malignant type of soft tissue sarcoma (STS) [1]. SS is relatively chemosensitive compared to other STS and neoadjuvant treatments were proven to improve survival [2]. Nonetheless, although local control has improved, metastases develop in 40% of patients, with lung involvement in the metastatic process in 90% of cases [3]. Therefore, surgery is considered a first-line treatment for metastatic SS [4] and it continues to evolve with the introduction of new techniques, including radiofrequency ablation (RA) and stereotactic body radiation therapy (SBRT) [5].

Chemotherapy is typically used as an adjuvant treatment but its proper role for high-grade SS (HGSS) remains unclear [6]. Therefore, we explored the survival benefit of adjuvant chemotherapy after first lung metastasectomy (LMTS) in HGSS patients. 

## 2. Material and Methods

### 2.1. Patient Population

Patients undergoing lung metastasectomy for HGSS at a tertiary STS referral center (Rizzoli Orthopedic Hospital, Bologna, Italy) between 2000 and 2020 were the subject of the study. As a result, HGSS was defined as grade ≥ 2 of the disease following the 2013 WHO/French Federation of Cancer Centers Sarcoma Group/National Cancer Institute (US) tumor grading system [7]. Moreover, all patients included had lung metastases, thus falling into stage IV of the 2017 classification by the American Joint Committee on Cancer (AJCC) [8].

Inclusion criteria were: (1) availability of histological examination for all primary tumors and resected nodules; (2) availability of clinical data related to the primary tumor; (3) available preoperative, postoperative, and follow-up CT images; and (4) no residual nodules at first metastasectomy. Patients also treated with RA and SBRT were excluded. Clinical data were recorded by one radiologist (C.T.), one clinician (M.R.), and one oncologist (A.L.). 

### 2.2. Definitions and Classifications

A pathologist reviewed all the histologic examinations. CT scan features were defined following the Fleischner Society Glossary of Terms for Thoracic Imaging [9,10]. The American Joint Committee on Cancer recommendations were followed for residual tumor R classification [11].

T_0_ was defined as the time at first LMTS, while T_1_–T_3_ referred to the time of further LMTS.

The primary endpoint was freedom from SS-specific mortality.

The disease-free interval at time zero (DFI_0_) was the interval between the diagnosis of the primary tumor and the first CT-diagnosed lung metastasis. The disease-free interval was calculated at any appearance of lung metastases and defined as the interval from the previous metastasectomy to the diagnosis of new-onset metastasis (DFI_1–4_).

First-line adjuvant chemotherapy was chosen based on DFI_0_. Ifosfamide and anthracyclines were the first choices and the primary tumor. Nonetheless, when the DFI_0_ was short, other regimens were applied.

### 2.3. Statistical Analysis

Data were tested for normality using the Shapiro–Wilk test and expressed as the mean ± SD or median and interquartile range. Statistical differences were analyzed using the Mann–Whitney test, whereas the other variables were analyzed with Pearson’s Χ^2^ test and Fisher’s exact test. Univariable and multivariable Cox analyses were used in the whole patient population to determine independent predictors of disease-specific mortality at follow-up. The proportional hazards assumption was checked for each covariate to test for independence between residuals and time. Additionally, a global test was performed for the model. The resulting p-values were not statistically significant, confirming the initial assumption.

All variables associated with *p* < 0.1 in the univariable analysis were entered into the multivariable Cox regression. Furthermore, to test for interaction terms, a subgroup analysis was performed to analyze the interactions between the main predictors. Cut-off values were found employing the receiver operating characteristic (ROC) curves. Survival analysis was performed using the Kaplan–Meier method, and the log-rank test was used to detect statistical differences between curves between the two groups. Finally, a sub-analysis was carried out comparing survival curves in the two groups by different variables. All analyses were corrected by the year of surgery. 

R software v. 3.5.3 (R Foundation for Statistical Computing, Vienna, Austria) was employed for analysis. A *p*-value of less than 0.05 was considered significant.

## 3. Results

### 3.1. Patient Population 

The patient population consisted of 46 HGSS patients divided into two groups, including (1) patients undergoing adjuvant chemotherapy after lung metastasectomy (*n* = 24) and (2) patients who did not receive chemotherapy following LMTS (*n* = 22). Table 1 shows the demographic characteristics and the primary tumor features in the two groups. No statistical difference was found between patients with or without adjuvant chemotherapy. Characteristics of metastases in the two groups are displayed in Table 2 and no significant difference was detected between the groups.

### 3.2. Chemotherapy

For the primary tumor, neoadjuvant/adjuvant chemotherapy was performed with a combination of Ifosfamide and Epirubicin ((*n* = 42) 9 g/m^2^ and 120 mg/m^2^, respectively) or Ifosfamide and Doxorubicin ((*n* = 4) 9 g/m^2^ and 75 mg/m^2^ (60 mg/m^2^ in patients older than 65 years), respectively) for each cycle. The number of cycles ranged from three to six applied every three weeks, with a treatment length comprised between 2 and 4 months. No significant difference was detected between the two study groups (*p* = 0.07). 

In 24 patients undergoing first-line adjuvant chemotherapy after lung metastasectomy, the following regimens were applied and administrated every 21 days starting from the first day: (1) high dose Ifosfamide (15 g/m^2^) in 5 days (*n* = 4); (2) Ifosfamide (9 g/m^2^ (6 g/m^2^ if >65 years), *n* = 8); (3) combination of Ifosfamide (9 g/m^2^ (6 g/m^2^ if >65 years)) and Epirubicin (120 mg/m^2^, *n* = 2); (4) Epirubicin (120 mg/m^2^, *n* = 2); (5) a combination of Gemcitabine (1800 mg/m^2^) and Docetaxel (75 mg/m^2^, *n* = 6); and (6) Trabectedin (1.3 mg/m^2^, *n* = 2).

The number of cycles ranged from four to five, with a treatment length comprised between 3 and 4 months.

In the case of disease progression and need for further chemotherapy lines, the following drugs were employed: (1) oral regimen of Pazopanib (800 mg/die (400 mg/die in the case of reduced renal/hepatic function)) and (2) Dacarbazine (850 mg/m^2^).

### 3.3. Survival Analysis

At a median follow-up of 32.5 months (IQR 18.0–82.7), thirty-two patients died of the disease, 21 were in the chemotherapy group, and 11 did not receive the treatment.

In patients undergoing adjuvant chemotherapy after LMTS, freedom from SS-specific mortality at 60 and 120 months was 20.8% (9.55–45.4%). In patients who did not receive the chemotherapy, this figure was significantly higher: 50.0 % (33.0–76.0%), *p* = 0.01 (Figure 1).

In the sub-analysis comparing actuarial curves in patients with or without chemotherapy by different variables, we found, in the chemotherapy group, significantly higher freedom from SS mortality in patients with mono-lateral metastases (Figure 2A). In contrast, we failed to find any difference in the group without chemotherapy (Figure 2B). Similarly, in the chemotherapy group (Figure 3A), patients with a primary tumor ≥ 10 cm showed a significantly lower survival (*p* < 0.001), whereas this difference was not found in patients without chemotherapy (Figure 3B, *p* = 0.9). No other differences were detected in the subgroups.

### 3.4. Chemotherapy as a Predictor of Death 

Table 3 displays the results of the univariable and multivariable Cox analysis. Chemotherapy was associated with a risk of death of 2.5 times higher (*p* = 0.014) at the univariable research stage. This was confirmed by multivariable Cox regression (HR 2.8, *p* = 0.02).

Likely, the presence of bilateral metastases significantly increased the risk of death at univariable and multivariable analysis (HR 2.4, *p* = 0.018 and HR 2.9, *p* = 0.0047, respectively). However, no other covariates resulted in being significant and among them, remarkably, the PT size was not a significant predictor of death.

To assess the independent value of chemotherapy as a predictor of survival compared to the other predictors, we carried out an interaction analysis. As displayed in Figure 4, chemotherapy as a predictor of death was not strengthened by the presence of bilateral metastases, which was the only significant co-predictor of death at multivariable analysis.

We further analyzed the covariate that, although not significant at Cox regression, could have strengthened the adverse effects of chemotherapy on death. Female sex increased the risk of dying of chemotherapy by more than six times. In addition, the HR was raised by approximately three times in patients older than 48 years by >five times when the primary tumor was > 10 cm and by slightly less than four times when the DFI_0_ was ≤6 months.

## 4. Discussion

Synovial sarcoma (SS) is a rare malignancy representing a soft tissue sarcoma (STS) of uncertain differentiation. It accounts for 5–10% of all STS [12,13]. It occurs at any age, even though young adults are mostly affected [14,15,16]. Lower extremities represent the most common primary site (70%), especially around the knee, but almost any anatomic site can be involved [17]. It occurs primarily in the para-articular regions, usually associated with tendon sheaths, bursae, and joint capsules [18]. Histologically, synovial sarcoma is characterized by epithelial-like and spindle cell components arranged in a biphasic or monophasic pattern. The classic biphasic type consists of spindle cells and epithelial cells (usually glandular structures). In contrast, the monophasic type comprises only spindle cells [17], which are a poorly differentiated type consisting of small cells. In addition, a poorly differentiated type consisting of cells resembling small round blue cell tumors has been described more recently [17,19]. 

Thus, the term synovial sarcoma has been used due to the similarity between cells of this tumor and primitive synoviocytes. However, it occurs in areas with no apparent relation to synovial structures [20]. Synovial sarcoma is considered an aggressive tumor and tends to give local recurrence as well as early and late metastases [14,21,22], and metastatic spread occurs in the lungs in approximately 70% [13].

The SS is considered relatively chemosensitive compared to other STS [23], with a response rate of 58.6% [24] compared to the 28–47% [25,26,27,28,29] in overall STS. Furthermore, the association between Ifosfamide and Doxorubicin improved survival in patients with advanced disease [24,30,31,32]. Nonetheless, these findings are based on small studies [25,26,33,34] and, more recently, the French Sarcoma Group showed no overall survival benefit with neoadjuvant or adjuvant chemotherapy in adults with SS [35].

However, there is little data regarding the role of adjuvant chemotherapy in treating LMTS in patients with HGSS [21]. In addition, adjuvant chemotherapy is often reserved for patients with advanced disease and for people whose all metastases cannot be entirely removed by surgery. Therefore, it is challenging to compare imbalanced populations and establish adjuvant chemotherapy’s effective influence in metastatic HGSS. The strength of our work is the balanced cohort between the two groups regarding either the primary tumor or lung metastases’ characteristics, as well as the exclusion of patients with incomplete first LMTS. These reduce the bias observed between patients chosen or not for chemotherapy and allow for a more reliable analysis of the effect of the adjuvant chemotherapy treatment on survival.

Moreover, our study is focused on a cohort of patients undergoing LMTS for HGSS over 20 years. There was no constant consensus regarding the most appropriate therapy regimen through such an extended period. Therefore, the indication for chemotherapy after the first metastasectomy followed the evolving institution policy: at the beginning, surgery was more aggressive, while in the following years, with the published evidence of the positive outcomes of chemotherapy in SS, there was an increasing use of adjuvant treatment in these patients. However, this was considered in the analysis corrected by the year of surgery.

Patients undergoing adjuvant chemotherapy after LMTS showed a 5-year freedom from SS-specific mortality significantly lower than patients not treated. In addition, chemotherapy raised the risk of death by 2.5 times during the follow-up. In our previous work [36], we demonstrated that chemotherapy negatively influences the recurrence of metastases. We postulated that chemotherapy might enhance metastasis recurrence by suppressing anti-tumor immunity [37,38]. The harmful effect in destroying the anti-tumor immune response and inhibiting the development of anti-tumor T-cell memory might favor the development of latent metastases, leading to recurrence [38]. Hence, it can be hypothesized that suppression of anti-tumor immunity might play a critical multifactorial role in worsening general patients’ clinical conditions leading to death. In our report, although the adverse effect of chemotherapy was also confirmed for survival, we failed to demonstrate that patients with chemotherapy had a higher number of lung recurrences. Indeed, after the first metastasectomy, more patients without chemotherapy had an oligometastatic lung spread, although this did not reach statistical significance.

Interestingly, in our experience, the negative effect of chemotherapy on survival was influenced neither by the bilateral occurrence of lung metastases nor by primary tumor size. Additionally, the number of patients with bilateral metastases at T_2_ and T_3_ was higher in subjects who underwent chemotherapy, although this did not reach statistical significance.

Finally, we analyzed factors that enhance the adverse effect of chemotherapy on disease-specific survival and found that DFI_0_ ≤ 6, female sex, age > 40 years, and primary tumor > 10 cm increased the risk of death by about four, six, >three, and >five times, respectively. In contrast, the presence of metastases in other sites, local recurrence, and adjuvant radiotherapy did not affect the influence of chemotherapy on the outcome.

## 5. Limitations

Our study has some limitations that must be highlighted. First, this concerns the retrospective nature of the research and the limited number of patients. Nonetheless, the rarity of STS makes a large sample size challenging to obtain and thus this limitation is shared with most published studies. Second, there are limitations regarding patients who were not amenable to surgical lung resection and only underwent chemotherapeutic treatment. This might have given more insight on the role of chemotherapy on lung metastatic HGSS. Third, the study was conducted long-term, implying different diagnostic and surgical techniques that have improved over time. However, the analysis was corrected by the year of surgery. Fourth, immune response-related blood data was not available. Finally, the paper does not include head SS, which are not referred to at our institution.

## 6. Conclusions

Adjuvant chemotherapy did not show any advantage in terms of disease-specific survival in HGSS patients undergoing metastasectomy. In contrast, it negatively correlated long-term disease-specific survival that was worsened by the female sex, age > 48 years, and primary tumor size > 10 cm. Prospective studies are needed to confirm our findings and explore whether chemotherapy might benefit advanced disease or when the disease-free interval is short.

## Figures and Tables

**Figure 1 jcm-10-05956-f001:**
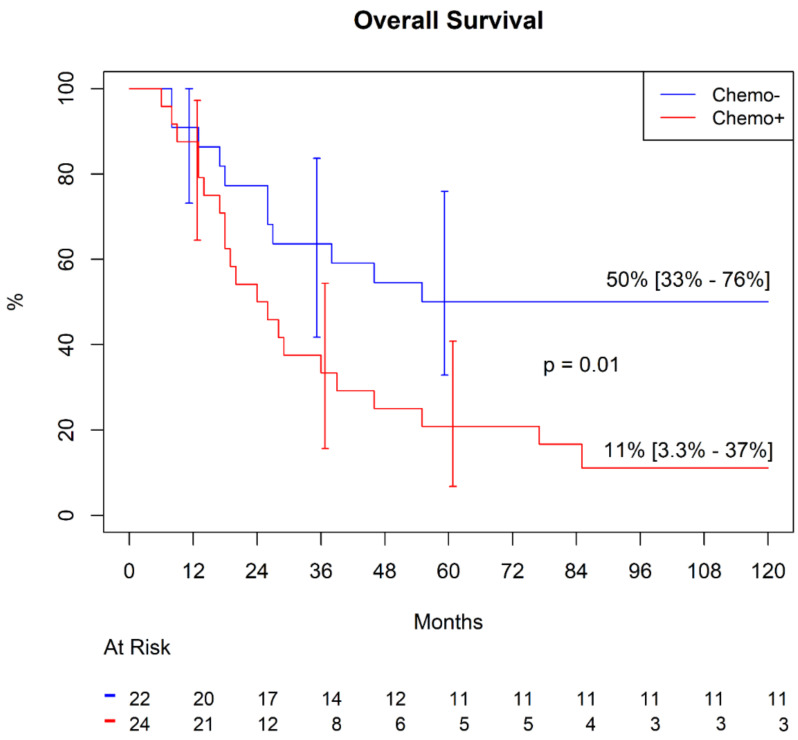
Freedom from synovial sarcoma (SS)-specific mortality.

**Figure 2 jcm-10-05956-f002:**
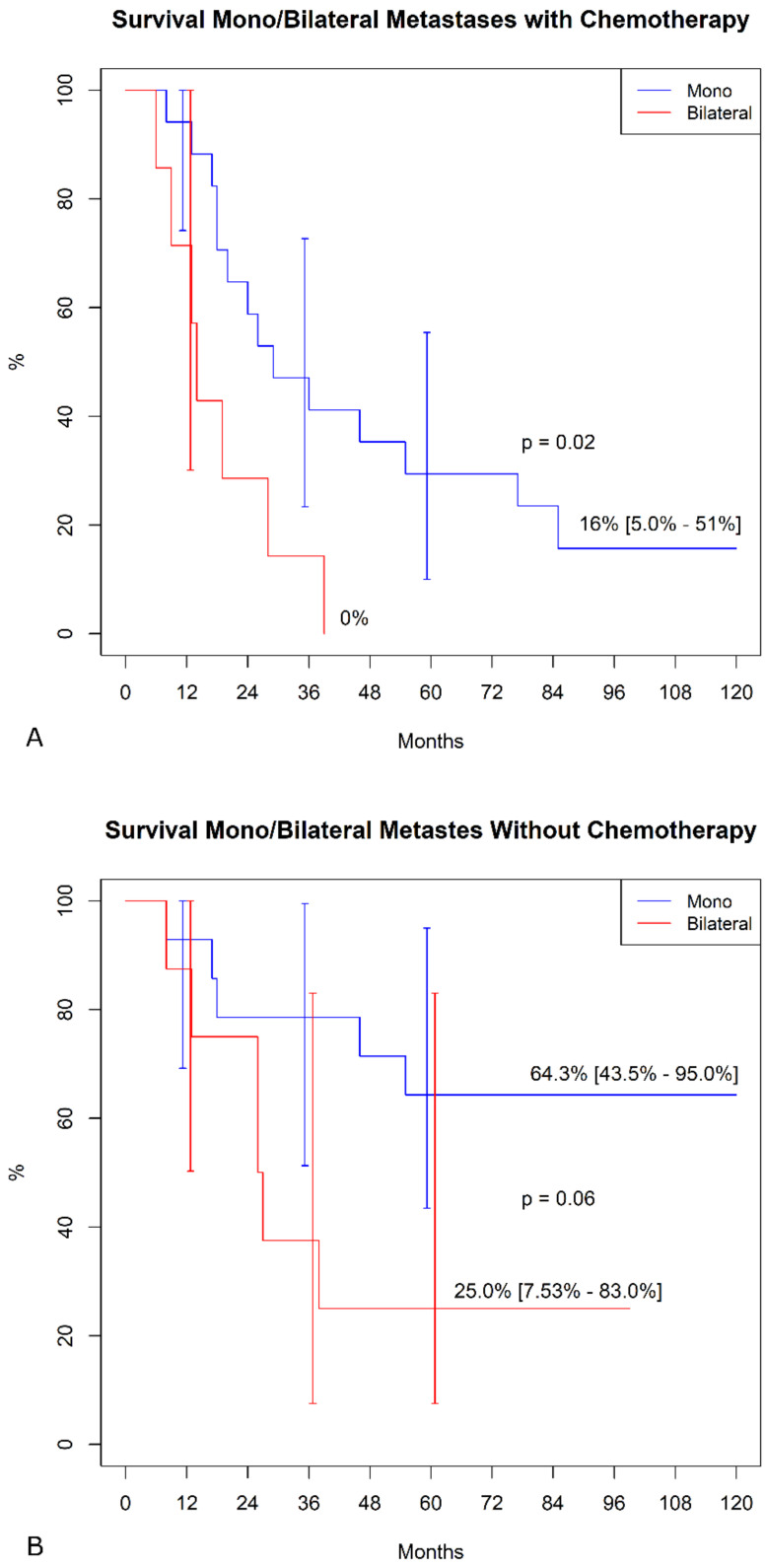
Freedom from SS-specific mortality in mono/bilateral metastasis. (**A**) Patients were undergoing adjuvant chemotherapy. (**B**)**.** Patient not treated with chemotherapy.

**Figure 3 jcm-10-05956-f003:**
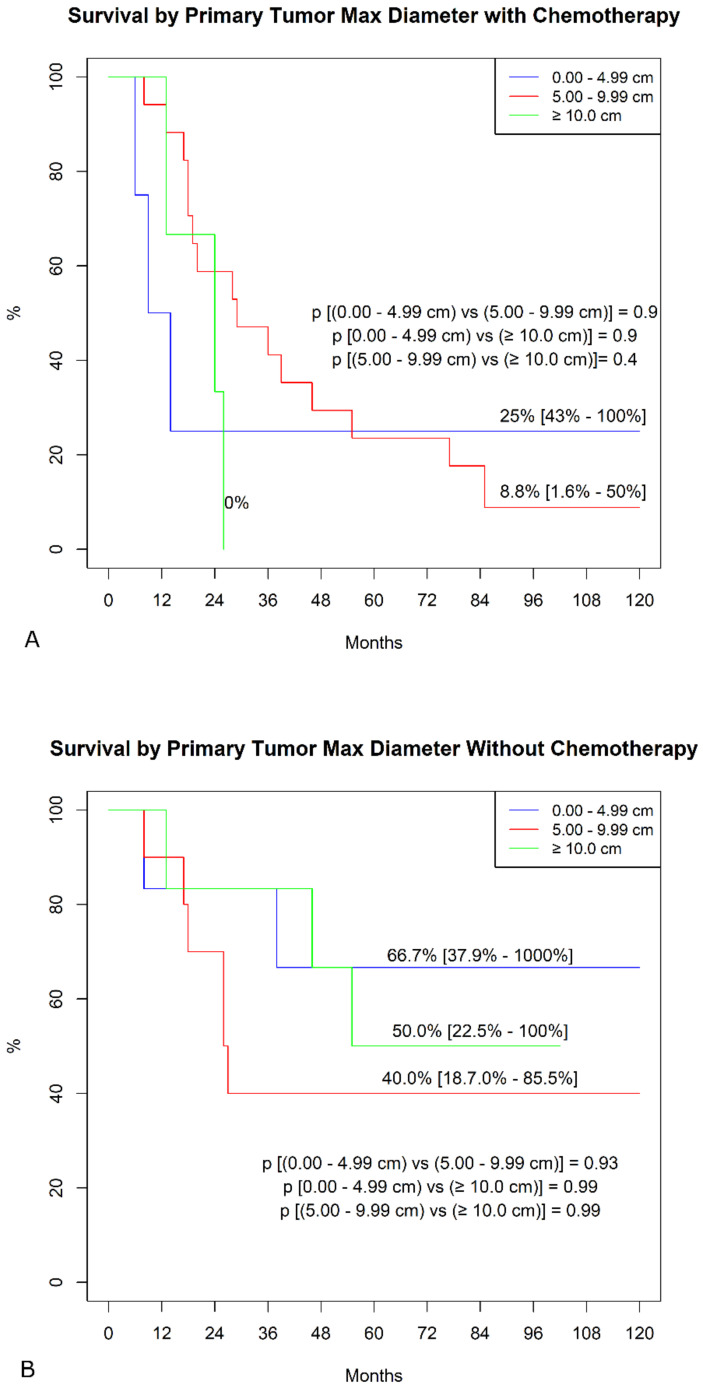
Freedom from SS-specific death by primary tumor maximal diameter. (**A**)**.** Patients were undergoing adjuvant chemotherapy. (**B**)**.** Patient not treated with chemotherapy.

**Figure 4 jcm-10-05956-f004:**
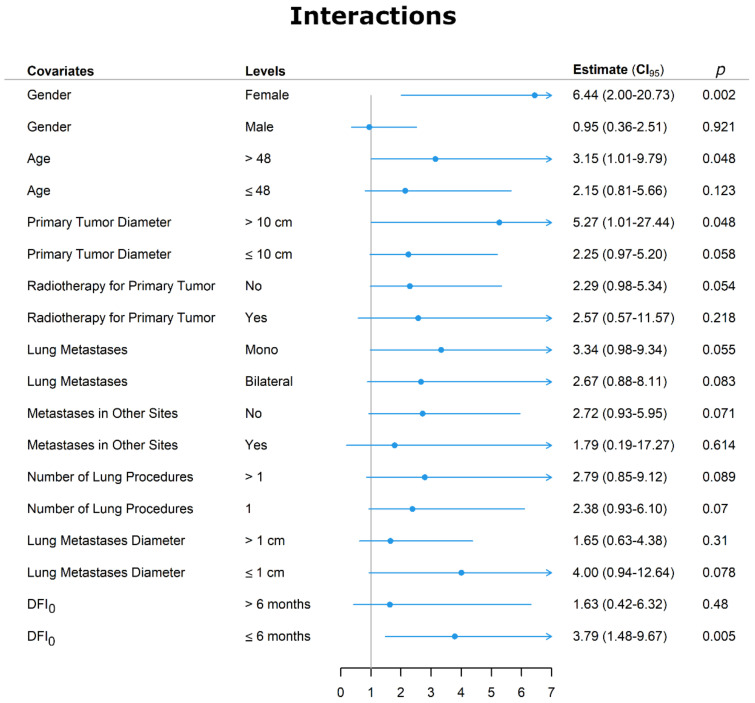
Interaction graph between chemotherapy and other variables. Abbreviations: DFI_0_ = disease-free interval at time zero was the interval between the diagnosis of the primary tumor and the first CT-diagnosed lung metastasis, and CI = confidence interval.

**Table 1 jcm-10-05956-t001:** Demographics and primary tumor characteristics (*n* = 46).

	Chemotherapy	No Chemotherapy	*p*
	*n* = 24	*n* = 22	
Age	43 ± 14	49 ± 14	0.12
Gender (female)	12 (50.0)	14 (63.6)	0.53
Primary tumor diameter (cm)			
0–5	6 (27.3)	4 (16.7)	
5–10	10 (45.4)	17 (70.8)	0.10
> 10	6 (27.3)	3 (12.5)	
Margins			
R0	15 (62.5)	11 (50.0)	
R1	6 (25.0)	8 (36.4)	0.66
R2	3 (12.5)	3 (13.6)	
Biphasic/Monophasic SS	3/21(12.5/87.5)	4/18 (18.2/81.8)	0.62
Primary tumor site			
Lower limbs	18 (75.0)	10 (45.5)	
Upper limbs	5 (20.8)	3 (13.6)	
Abdominal wall	0 (0.0)	1 (4.5)	0.08
Back	0 (0.0)	3 (13.6)	
Neck	0 (0.0)	2 (9.1)	
Gluteus	1 (4.2)	3 (13.6)	
Limbs	23 (95.8)	13 (60.0)	0.003
Trunk	1 (4.1)	7 (46.6)	0.01
Neck	0 (0)	2 (9.1)	0.22
Primary tumor chemotherapy			
Overall chemotherapy yes/no	20/4 (83.3/16.7)	19/3 (86.4/13.6)	0.79
Neoadjuvant	3 (12.5)	9 (40.9)	0.08
Adjuvant	9 (37.5)	8 (36.4)	
Combined	8 (33.3)	2 (9.1)	
Primary tumor radiotherapy	6 (26.1)	8 (36.4)	0.67
DFI_0_	6.00 [5.00, 9.00]	5.00 [4.00, 7.00]	0.38

Data expressed as median (interquartile range) or number (percentages). Abbreviations: DFI_0_ = disease-free interval from the primary tumor resection to the first diagnosed metastasis.

**Table 2 jcm-10-05956-t002:** Characteristics of lung metastases (*n* = 46).

	Chemotherapy *n* = 24	No Chemotherapy *n* = 22	*p*
Number of LM at T_0_			
1–4	15 (62.5)	18 (81.8)	
5–10	7 (29.1)	2 (9.1)	0.22
>10	2 (8.3)	2 (9.1)	
Bilateral LM	7 (29.2)	8 (36.4)	0.84
Max diameter of LM	1.15 [0.95–2.00]	1.05 [0.83–1.48]	0.40
Number of LM at T_0_			
Biphasic	5.5 [2,3,4,5,6,7,8,9]	4.5 [1,2,3,4,5,6,7,8]	0.6
Monophasic	2 [1,2,3]	2 [1,2,3,4]	0.4
Metastases at other sites	4 (16.7)	2 (9.1)	0.75
PT local recurrence	9 (37.5)	9 (40.9)	>0.9
Surgery			
Wedge	23 (95.8)	21 (95.5)	
Segmentectomy	1 (4.2)	0 (0.0)	0.37
Lobectomy	0 (0.0)	1 (4.5)	
LM among survivors at T_1_			
1–4	9 (37.5)	10 (47.6)	
5–10	8 (33.3)	1(4.7)	0.09
>10	7 (29.2)	3 (14.3)	
LM among survivors at T_2_			
1–4	5 (45.5)	9 (75.0)	
5–10	4 (36.3)	2 (16.6)	0.34
>10	2 (18.1)	1 (8.3)	
DFI_1_	4.00 [3.00–7.00]	4.00 [2.50–4.00]	0.48
DFI_2_	10.33 [7.64–14.2)	12.00 [7.21–15.4]	0.80
Inoperable after first LMTS	11 (45.8)	9 (40.9)	>0.9

Data expressed as median (interquartile range) or number (percentages). Abbreviations: LM = lung metastases; DFI = disease-free interval from the previous metastasectomy to the diagnosis of new-onset metastasis (DFI_1-2_); LMTS = lung metastasectomy; T_0_: time at first LMTS; and T_1_–T_3:_ at first LMTS at second and third LMTS (if applicable).

**Table 3 jcm-10-05956-t003:** Cox analysis.

Covariate	Univariate			Multivariate		
	HR	95% CI	*p*	HR	95% CI	*p*
Chemotherapy	2.5	1.2–5.2	0.014	2.8	1.2–6.6	0.02
PT size	0.84	0.67–1	0.1	0.96	0.74–1.2	0.73
LM Mono/Bilateral	2.4	1.2–4.9	0.018	2.9	1.4–6.3	0.0047

Abbreviations: HR = hazard ratio; CI = confidence interval; PT = primary tumor; and LM = lung metastases.

## Data Availability

Data available on reasonable request.

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
