# Peer review of "Chemosensitivity of Lung Metastatic High-Grade Synovial Sarcoma"

_jcm, 2021, doi:10.3390/jcm10245956_

Round 1
Reviewer 1 Report
Thank you for your manuscript report adjuvant chemotherapy did not show any advantage in terms of survival in HGSS patients.
Author Response
Thank you for your manuscript report adjuvant chemotherapy did not show any advantage in terms of survival in HGSS patients.
First, we would like to thank this reviewer for the positive comments
- English language and style are fine/minor spell check required
Reply: As requested by this and the other reviewers, the language was re-checked by of Academic & Scientific Editing Services for the English editing of the paper.
Changes: Accordingly.
- Improve: Are the methods adequately described?
Reply: Methods have been implemented also according to the requests of all reviewers
Changes:
In the methods we added:
First-line adjuvant chemotherapy was chosen based on DFI0. Ifosfamide and anthracyclines were the first choices as well as for the primary tumor. Nonetheless, when the DFI0 was short other regiments were applied.
The main endpoint was freedom from SS-specific mortality.
Moreover, all patients included had lung metastases, thus falling into Stage IV of 2017 classification by the American Joint Committee on Cancer (AJCC) (8).
Inclusion criteria were: 1) Availability of histological examination for all primary tumors.
Tables 1 and 2 were implemented
- Improve: Are the conclusions supported by the results?
Reply: Conclusions have been implemented also according to the requests of all reviewers
Changes:
Conclusions were changed to:
Adjuvant chemotherapy did not show any advantage in terms of disease-specific survival in HGSS patients undergoing metastasectomy. In contrast, it negatively correlated with long-term disease-specific survival that was worsened by the female sex, age > 48 years, and primary tumor size >10 cm. Further prospective studies are needed to confirm our findings and to explore whether chemo-therapy might have a benefit in case of oligometastatic disease or when the disease-free interval is short.
Reviewer 2 Report
The article deals with a very important topic, namely do patients with synovial sarcoma after complete metastasectomy benefit from adjuvant chemotherapy.
However, there are some unclarities to resolve:
- Chemotherapy as such is viewed/described very globally. But the treating oncologists are interested in: which substances were given. How many cycles were recommended /given. Chemotherapy is not just one substance, and the cumulative doses/ length of therapy might be interesting too. Since this is a single centre evaluation, this information should be available. This information is an extremely important information for the treating physicians. Your report will gain importance if you can clearly specify which substancs were not effective.
- REsults: 3.1. Patient Population: I doubt the results in table Table 1. Demographics and Primary Tumor characteristics (n=46). The classification of primary tumor site and primary tumor chemotherapy is very granular. This limits statistical power. So, I would propose to add additionally (do not delete anything!!!, but add):
- Primary Tumor Site: limb versus trunk versus head/neck
- Primary chemotherapy: Chemotherapy versus no chemotherapy
- With regard to: 3.2. Survival Analysis
"At a median follow up of 32.5 [IQR 18.0-82.7]": which unit: months?
- Discussion: Here is something written twice: "However, there is little data regarding the role of adjuvant chemotherapy in treating
LMTS in patients with HGSS (21). In addition, adjuvant chemotherapy is often reserved
for patients with advanced disease and for people whose all metastases cannot be entirely
removed by surgery. This makes it challenging to compare imbalanced populations and
establish adjuvant chemotherapy's effective influence in metastatic HGSS. The strength of
our work is the balanced cohort between the two Groups regarding either the primary
tumor or lung metastases' characteristics and the exclusion of patients with incomplete
first LMTS. These reduce the bias observed between patients chosen or not for chemotherapy and allow a more reliable analysis of the effect of the adjuvant chemotherapy treatment on survival. In addition, adjuvant chemotherapy is often reserved for patients with
advanced disease and for people for people whose all metastases cannot be completely
removed by surgery. This makes it difficult to compare imbalanced populations and to
establish the effective influence of adjuvant chemotherapy in metastatic HGSS. The
strength of our work are the balanced cohort between the two Groups regarding either
the primary tumor or lung metastases’ characteristics, as well as the exclusion of patients
with incomplete first LMTS. These reduce the bias that frequently is observed between
patients chosen or not for chemotherapy and allows a more reliable analysis of the effect
of the adjuvant chemotherapy treatment on survival." - Again something is somehow doubled: "Moreover, our study is focused on a cohort of patients undergoing LMTS for HGSS
over 20 years. Through such a long-time span, there was no constant consensus regarding
the most appropriate therapy regimen. Therefore, the indication for chemotherapy after
the first metastasectomy followed the evolving Institution policy: at the beginning, surgery was more aggressive, while in the following years, with the published evidence of
positive outcomes of chemotherapy in SS, there was an increasing use of adjuvant treatment in these patients. However, this was taken into account in the analysis that was corrected by the year of surgery. there was no constant consensus regarding the most appropriate therapy regimen. Therefore, the indication for chemotherapy after the first metastasectomy followed the evolving Institution policy: at the beginning, surgery was more
aggressive while in the following years, with the published evidence of positive outcomes
of chemotherapy in SS, there was an increasing use of adjuvant treatment in these patients.
However, this was taken into account in the analysis that was corrected by the year of
surgery" - the authors have made very careful and very elaborate compilations of the disease-free intervals. this was certainly a lot of work, which I would like to acknowledge herewith. However, the conclusion refers exclusively to the overall survival. And here the endpoint is death. As the result is that "Adjuvant chemotherapy did not show any advantage in terms of survival in HGSS
patients undergoing metastasectomy. In contrast, it had a negative impact on long-term
survival .... " everybody asks what kind of survival? What was the cause of death (complications of chemotherapy)? what was the disease-specific mortality? Did they all die of their tumor or chemotherapy-associated complications? - This is not a randomized trial but a retrospective analysis, so the conclusion is too strongly worded for that:
I would propose to rephrase with "In contrast --" it correlated with... - "Further larger studies..." I think that prospective studies are needed, not only larger
- It is disappointing for the reader to have a negative result at the end. It should also be stated here in which situation chemotherapy might have a small benefit - and if so, which benefit - or in which situation it might have a small benefit. (for further studies). This also reinforces the negative statement of the entire work.
Author Response
The article deals with a very important topic, namely do patients with synovial sarcoma after complete metastasectomy benefit from adjuvant chemotherapy.
Thank this reviewer for her/his insightful suggestion and nice comments.
The language was re-checked by of Academic & Scientific Editing Services for the English editing of the paper.
- Chemotherapy as such is viewed/described very globally. But the treating oncologists are interested in which substances were given. How many cycles were recommended /given.Chemotherapy is not just one substance, and the cumulative doses/ length of therapy might be interesting too. Since this is a single center evaluation, this information should be available. This information is an extremely important information for the treating physicians. Your report will gain importance if you can clearly specify which substances were not effective.
Reply: We agree and give available information.
Regarding the drug efficacy, it also depends on the time of administration. Most of our patients received similar regimens as first line chemotherapy. In patients received more than 1 line because of the progression of the disease the drugs were less effective.
Changes: A new subheading Chemotherapy was added:
“3.3 Chemotherapy
For the primary tumor neoadjuvant/adjuvant chemotherapy was performed with a combination of Ifosfamide and Epirubicin ([n=42] 9g/m2 and 120 mg/m2, respectively) or Ifosfamide and Doxorubicin ([n=4] 9g/m2 and 75 mg/m2 [60 mg/m2 in patients older than 65 yrs.], respectively) for each cycle. The number of cycles ranged from 3 to 6 applied every 3 weeks, with a treatment length comprised between 2 and 4 months. No significant difference was detected between the two study groups (p=0.07).
In 24 patients undergoing first-line adjuvant chemotherapy after lung metastasectomy, the following regimens were applied, administrated every 21 days starting from the first day: 1) High dose Ifosfamide (15g/m2) in 5 days (n=4). 2) Ifosfamide (9g/m2 [6g/m2 if >65 years], n=8). 3) Combination of Ifosfamide (9g/m2 [6g/m2 if >65 years]) and Epirubicin (120 mg/m2, n=2). 4) Epirubicin (120 mg/m2), (n=2). 5) A combination of Gemcitabine (1800 mg/m2) and Docetaxel (75 mg/m2), (n=6). 6) Trabectedin (1.3 mg/m2, n=2).
The number of cycles ranged from 4 to with a treatment length comprised between 3 and 4 months.
In case of disease progression and need for further chemotherapy lines, the following drugs were employed: 1) Oral regimen of Pazopanib (800 mg/die [400 mg/die in case of reduced renal /hepatic function]). 2) Dacarbazine (850 mg/m2).”
In the methods we added:
“First-line adjuvant chemotherapy was chosen based on DFI0. Ifosfamide and an-thracyclines were the first choices as well as for the primary tumor. Nonetheless, when the DFI0 was short other regiments were applied.”
Results: 3.1. Patient Population: I doubt the results in table Table 1. Demographics and Primary Tumor characteristics (n=46). The classification of primary tumor site and primary tumor chemotherapy is very granular. This limits statistical power. So, I would propose to add additionally (do not delete anything!!!, but add): Primary Tumor Site: limb versus trunk versus head/neck Primary chemotherapy: Chemotherapy versus no chemotherapy
Reply: We agree with this reviewer and add the information in Table 1 accordingly.
Head SS are not referred to our center therefore this data could not be included. This was added to the limitations section
Changes:
Table 1: information requested was added.
Limitation section “ Finally, the paper does not include head SS which are not referred to our Institution” was added.
- With regard to: 3.2. Survival Analysis "At a median follow up of 32.5 [IQR 18.0-82.7]": which unit: months?
Reply: The interpretation is correct ( see also reviewer 3). Sorry for missing it.
Changes: “ months” was added in section 3.2 line 1.
- Discussion: Here is something written twice: "However, there is little data regarding the role of adjuvant chemotherapy in treating LMTS in patients with HGSS (21). In addition, adjuvant chemotherapy is often reserved for patients with advanced disease and for people whose all metastases cannot be entirely removed by surgery. This makes it challenging to compare imbalanced populations and establish adjuvant chemotherapy's effective influence in metastatic HGSS. The strength of our work is the balanced cohort between the two Groups regarding either the primary tumor or lung metastases' characteristics and the exclusion of patients with incomplete first LMTS. These reduce the bias observed between patients chosen or not for chemotherapy and allow a more reliable analysis of the effect of the adjuvant chemotherapy treatment on survival.
In addition, adjuvant chemotherapy is often reserved for patients with advanced disease and for people for people whose all metastases cannot be completely removed by surgery. This makes it difficult to compare imbalanced populations and to establish the effective influence of adjuvant chemotherapy in metastatic HGSS. The strength of our work are the balanced cohort between the two Groups regarding either the primary tumor or lung metastases’ characteristics, as well as the exclusion of patients with incomplete first LMTS. These reduce the bias that frequently is observed between patients chosen or not for chemotherapy and allows a more reliable analysis of the effect of the adjuvant chemotherapy treatment on survival."
Reply: Thank you for underlining this. Sorry for such a mistake. Changes were made accordingly.
Changes: The duplicated text was removed.
- Again something is somehow doubled: "Moreover, our study is focused on a cohort of patients undergoing LMTS for HGSSover 20 years. Through such a long-time span, there was no constant consensus regarding the most appropriate therapy regimen.
Therefore, the indication for chemotherapy after the first metastasectomy followed the evolving Institution policy: at the beginning, surgery was more aggressive, while in the following years, with the published evidence of positive outcomes of chemotherapy in SS, there was an increasing use of adjuvant treatment in these patients. However, this was taken into account in the analysis that was corrected by the year of surgery. there was no constant consensus regarding the most appropriate therapy regimen.
Therefore, the indication for chemotherapy after the first metastasectomy followed the evolving Institution policy: at the beginning, surgery was more aggressive while in the following years, with the published evidence of positive outcomes of chemotherapy in SS, there was an increasing use of adjuvant treatment in these patients. However, this was taken into account in the analysis that was corrected by the year of surgery"
Reply: Thank you for underlining this. Sorry for such a mistake. Changes were made accordingly.
Changes: The duplicated text was removed.
- The authors have made very careful and very elaborate compilations of the disease-free intervals. this was certainly a lot of work, which I would like to acknowledge herewith. However, the conclusion refers exclusively to the overall survival. And here the endpoint is death. As the result is that "Adjuvant chemotherapy did not show any advantage in terms of survival in HGSS patients undergoing metastasectomy. In contrast, it had a negative impact on long-term survival .... " everybody asks what kind of survival? What was the cause of death (complications of chemotherapy)? what was the disease-specific mortality? Did they all die of their tumor or chemotherapy-associated complications?
Reply: We thank this reviewer for such an insightful observation. We agree that the endpoint was not clear for potential readers.
All patients were in an advanced stage of the disease since our research exluded early stages. All patients died of SS therefore we referred to disease specific mortality/ survival
Changes: The endpoint was better defines as freedom from SS-specific mortality. Changes were made throughout the manuscript.
- This is not a randomized trial but a retrospective analysis, so the conclusion is too strongly worded for that: I would propose to rephrase with "In contrast --" it correlated with... "Further larger studies..." I think that prospective studies are needed, not only larger It is disappointing for the reader to have a negative result at the end. It should also be stated here in which situation chemotherapy might have a small benefit - and if so, which benefit - or in which situation it might have a small benefit. (for further studies). This also reinforces the negative statement of the entire work.
Reply: We fully agree. Thank you for the helpful advices.
Changes: The conclusions were changed to:
“….In contrast, it negatively correlated had a negative impact on with long-term disease-specific survival. …Prospective studies are needed to confirm our findings and to explore whether chemo-therapy might have a benefit in case of oligometastatic disease or when the disease-free interval is short.
Reviewer 3 Report
Major corrections
- Methods: Stage or Grade? Please, use Arabic numerals for Grade (possible classyfications of Grade: WHO/ French Federation of Cancer Centers Sarcoma Group/ National Cancer Institute (US) tumor grading system) and Roman numerals for Stage (AJCC classyfication). Plesease, correct the sentence: "HGSS was defined as Grade ≥ GII of the disease, following the new classification by the American Joint Committee on Cancer (AJCC)." and add the year of AJCC in text.
- Plesease, describe the chemotherapy (ChT; names of medications) which was given to your patients. Why some of patients undergo ChT and other not?
- Methink, histological data should be placed not at the beginning of discussion but in methods chapter.
- Usually epithelial component of SS metastases. Were metastases to lungs/ recurrences components (epithelial/mesenchymal; if biphasic SS) similar in groups with/without chemotherapy?
- "We had postulated that chemotherapy might enhance metastasis recurrence by suppressing anti-tumor immunity." - do you have patients complete blood count?
Minor corrections
- The lines in graphs are too thin and have no colour. The legends are have too small letters.
- Lack of: Authors contributions (who from authors examined histologically tumors? - "A pathologist reviewed all the histologic examinations, and the grading system was updated following the French Federation of Cancer Centers Sarcoma Group classification (8)."), Institutional Review Board Statement and Informed Consent Statement - in correct position at the end of the article.
- Presence of the double negation: "Second, we could not include patients not amenable for surgical lung resection".
- Some of editorial corrections are needed: full stops, upper case/lowercase letters, spaces (examples: " between 2000-2020We divided them into two groups: 1) undergoing adjuvant chemotherapy (n=24). 2) Not receiving adjuvant chemotherapy (n=22)."; "disease-Free "; "T1-T3 referred"; "Inclusion criteria were: 1) Availability of..."; "... follow-up CT images. 4) No residual nodules..."; "It accounts for 5–10% of all STS(12, 13)"; new paragraphs after first sentences in introduction and discussion).
- Lack of "months" in sentence: "At a median follow up of 32.5 [IQR 18.0-82.7], thirty-two patients died, 21 belonging to the chemotherapy group and 11 who did not receive the treatment."
- Repetitions in discussion: "Nonetheless, these findings are based on small studies (25, 26, 33, 34) and, more recently, the French Sarcoma Group showed no overall survival benefit with neoadjuvant or adjuvant chemotherapy in adults with SS (35). with SS (35)." and "..., although this did not reach statistical significance. and T3 was higher in subjects who underwent chemotherapy although this did no not reach statistical significance.".
Author Response
We are Grateful to this reviewer for the precious suggestions.
The language was re-checked by of Academic & Scientific Editing Services for the English editing of the paper.
Major corrections
- Methods: Stage or Grade? Please, use Arabic numerals for Grade (possible classyfications of Grade: WHO/ French Federation of Cancer Centers Sarcoma Group/ National Cancer Institute (US) tumor grading system) and Roman numerals for Stage (AJCC classyfication). Plesease, correct the sentence: "HGSS was defined as Grade ≥ GII of the disease, following the new classification by the American Joint Committee on Cancer (AJCC)." and add the year of AJCC in text.
Reply: Agree. This was confusing.
Changes: we clarified as it follows in 2.1: HGSS was defined as Grade ≥ 2 of the disease, following the 2013 WHO/ French Federation of Cancer Centers Sarcoma Group/ National Cancer Institute (US) tumor grading system (7).
Moreover, all patients included had lung metastases, thus falling into Stage IV of 2017 classification by the American Joint Committee on Cancer (AJCC) (8).
- Plesease, describe the chemotherapy (ChT; names of medications) which was given to your patients. Why some of patients undergo ChT and other not?
Reply: We agree and give available information.
Changes: A new subheading Chemotherapy was added:
“3.3 Chemotherapy
For the primary tumor neoadjuvant/adjuvant chemotherapy was performed with a combination of Ifosfamide and Epirubicin ([n=42] 9g/m2 and 120 mg/m2, respectively) or Ifosfamide and Doxorubicin ([n=4] 9g/m2 and 75 mg/m2 [60 mg/m2 in patients older than 65 yrs.], respectively) for each cycle. The number of cycles ranged from 3 to 6 applied every 3 weeks, with a treatment length comprised between 2 and 4 months. No significant difference was detected between the two study groups (p=0.07).
In 24 patients undergoing first-line adjuvant chemotherapy after lung metastasectomy, the following regimens were applied, administrated every 21 days starting from the first day: 1) High dose Ifosfamide (15g/m2) in 5 days (n=4). 2) Ifosfamide (9g/m2 [6g/m2 if >65 years], n=8). 3) Combination of Ifosfamide (9g/m2 [6g/m2 if >65 years]) and Epirubicin (120 mg/m2, n=2). 4) Epirubicin (120 mg/m2), (n=2). 5) A combination of Gemcitabine (1800 mg/m2) and Docetaxel (75 mg/m2), (n=6). 6) Trabectedin (1.3 mg/m2, n=2).
The number of cycles ranged from 4 to with a treatment length comprised between 3 and 4 months.
In case of disease progression and need for further chemotherapy lines, the following drugs were employed: 1) Oral regimen of Pazopanib (800 mg/die [400 mg/die in case of reduced renal /hepatic function]). 2) Dacarbazine (850 mg/m2).”
In the methods we added:
“First-line adjuvant chemotherapy was chosen based on DFI0. Ifosfamide and an-thracyclines were the first choices as well as for the primary tumor. Nonetheless, when the DFI0 was short other regiments were applied.”
- Methink, histological data should be placed not at the beginning of discussion but in methods chapter.
Reply: We respectfully underline that no clinical data referring to our patiente were reported into the discussion. There is a literature overview about histology. However, as requested by this reviewer, we added some histological data in Table 1.
Changes: Hystological data added In table 1.
- Usually epithelial component of SS metastases. Were metastases to lungs/ recurrences components (epithelial/mesenchymal; if biphasic SS) similar in groups with/without chemotherapy?
Reply: Agree.
Changes: Number of metastases in Mono/biphasic was added in Table 2, whereas number of epithelial/mesenchymal biphasic and number of monophasic was specified in Table 1. No difference between YES/NO Chemotheraphy.
- "We had postulated that chemotherapy might enhance metastasis recurrence by suppressinganti-tumor immunity." - do you have patients complete blood count?
Reply: We respectfully highlight that this was postulated in our cited previous work and based on comparable results of Gao et. al and Shibayama et al.
Immune-response-related blood data was not available.
Changes: In the Limitation section: “Immune-response-related blood data was not available” was added.
Minor corrections
- The lines in graphs are too thin and have no colour. The legends are have too small letters.
Reply: We agree.
Changes: All figures were replaced and re-made following the reviewer’s suggestions.
- Lack of: Authors contributions (who from authors examined histologically tumors? - "A pathologist reviewed all the histologic examinations, and the grading system was updated following the French Federation of Cancer Centers Sarcoma Group classification (8)."), Institutional Review Board Statement and Informed Consent Statement - in correct position at the end of the article.
Reply: The pathologist who have examined the over time the specimens was added to the Aknowledgments. He gave his consent for being acknowledged.
Changes: Aknowledgments. Our heartfelt thanks to Dr Marco Gambarotti for pathologic examination.” Was added. Ethycal issues moved to the end of the article as requested.
- Presence of the double negation:"Second, we could not include patients not amenable for surgical lung resection".
Reply: We agree. We made the sentence clearer.
Changes: Corrected.
- Some of editorial corrections are needed: full stops, upper case/lowercase letters, spaces (examples: " between 2000-2020We divided them into two groups: 1) undergoing adjuvant chemotherapy (n=24). 2) Not receiving adjuvant chemotherapy (n=22)."; "disease-Free "; "T1-T3 referred"; "Inclusion criteria were: 1) Availability of..."; "... follow-up CT images. 4) No residual nodules..."; "It accounts for 5–10% of all STS(12, 13)"; new paragraphs after first sentences in introduction and discussion).
Reply: Thank you for the suggestion
Changes: Made.
- Lack of "months" in sentence: "At a median follow up of 32.5 [IQR 18.0-82.7], thirty-twopatients died, 21 belonging to the chemotherapy group and 11 who did not receive the treatment."
Reply: The interpretation is correct ( see also review 2), Sorry for missing it.
Changes: “ months” was added.
- Repetitions in discussion: "Nonetheless, these findings are based on small studies (25, 26, 33, 34) and, more recently, the French Sarcoma Group showed no overall survival benefit with neoadjuvant or adjuvant chemotherapy in adults with SS (35). with SS (35)." and "...,although this did not reach statistical significance. and T3 was higher in subjects who underwent chemotherapy although this did no not reach statistical significance.".
Reply: Thank you for underlining this. Sorry for the mistakes. Changes were made accordingly.
Changes: Repeated sentences were deleted.